# Correlations between Morphology, the Functional Properties of Upper Airways, and the Severity of Sleep Apnea

**DOI:** 10.3390/jcm11185347

**Published:** 2022-09-12

**Authors:** Adriana Neagos, Mihai Dumitru, Cristian Mircea Neagos, Mihaela Mitroi, Daniela Vrinceanu

**Affiliations:** 1ENT Department, George Emil Palade University of Medicine, Pharmacy, Science and Technology of Targu Mures, 540142 Tirgu Mures, Romania; 2ENT Department, Carol Davila University of Medicine and Pharmacy, 010271 Bucharest, Romania; 3Faculty of Medicine, Carol Davila University of Medicine and Pharmacy, 010271 Bucharest, Romania; 4ENT Department, University of Medicine and Pharmacy, 200349 Craiova, Romania

**Keywords:** obstructive sleep apnea, cephalometric, severity

## Abstract

Obstructive sleep apnea (OSA) is considered the silent killer pathology of the new millennium. This is due to increased risk factors such as obesity. Healthcare systems face an increasing burden from severe cases of OSA. We performed a study on a group of 152 Romanian patients with OSA recording data obtained through polysomnography and cephalometric variables, recorded in lateral plain X-rays. The results confirmed some of the data available from previous studies worldwide, but some of the variables presented a positive statistical correlation specific to our study group. For example, the apnea-hypopnea index (AHI) correlated with the uvula length but surprisingly did not correlate with body mass index (BMI) because obesity tends to become endemic in Romania. To our knowledge, this is one of the first studies focusing on cephalometric data in Romanian OSA patients. The results obtained through this study will be further analyzed in research on larger groups of Romanian OSA patients.

## 1. Introduction

Obstructive sleep apnea (OSA) is a major sleep disorder with a male prevalence of 20% and up to 15% in females. In many cases this is underdiagnosed, and prolonged exposure of the human body to hypoxia leads to cardiovascular-associated pathology. In developing countries, this pathology remains neglected in up to 90% of the population [1].

OSA combines symptoms such as snoring, sleep deprivation due to the phenomenon of apnea-hypopnea, and hypercapnia due to inadequate ventilation [2].

Therefore, OSA leads to numerous complications with a clear effect on cardiovascular mortality and morbidity, neurological complications, and altered metabolism. OSA is centered on anatomical and functional abnormalities leading to a partial neuromuscular collapse in upper airways during sleep. The main sites of obstruction are the nasal cavity and nasopharynx, soft palate, and tongue base, but more often the obstruction is multi-level [3].

Adenoid hypertrophy is responsible for sleep apnea in pediatric cases and leads to a vicious circle of associated pathologies: chronic sinusitis, recurrent otitis media, and changes in teeth development [4]. Another cause may be cranio-facial malformations [5]. Moreover, it is crucial that clinicians understand the considerable phenotypic variance in pediatric obstructive sleep apnea syndrome, as described by Tan HL, et al. [6]. In adults, obesity represents a high-risk factor for obstructive sleep apnea along with other specific body traits like cranio-facial morphology. There have been some studies focusing on correlating the skull bone measurements, obtained through plain X-rays, and clinical scales such as the AHI apnea-hypopnea index; therefore, a maxillary bone reduced in dimensions, a modified body of the mandible, an inferior positioning of the hyoid bone, and retrognathia are all associated with OSA due to a diminished volume in the viscerocranium [7].

In a very recent study by Bertuzzi F., et al. on 30 patients, it was underlined the fact that in normal-weight subjects, hard tissue-related factors have a greater impact on OSA severity, whereas, in overweight subjects, the impact of fat tissue is greater [8].

One possible site of obstruction of the airflow, and thus the origin of OSA in many cases, is represented by pathology at the level of the basal cavity: deviated nasal septum, enlarged turbinates or adenoids, and these pathologic states should be investigated in all cases presenting OSA [9].

Moreover, another possible site of obstruction is the pharynx. In this anatomic setting, an increase in soft tissue laxity reduces the functionality of the airway [10]. There are three narrowings: the space behind the palate, the lumen behind the tongue base, and the virtual space behind the epiglottis [11].

Macroglossia induced by obesity or retrognathia permits the advancement of the tongue base in the virtual lumen of the hypopharynx. This is a more common scenario for sleep apnea, unfortunately [12].

Another aspect involved in OSA is the pulmonary hypoxic vasoconstriction, which acts as a compensating mechanism for distributing the blood flow from hypoxic areas towards more oxygenated pulmonary tissue. The stimulus for activating this mechanism is a decrease in the partial pressure of alveolar oxygen due to hypoventilation [13,14,15,16].

In OSA, pulmonary hypertension develops due to a left ventricle dysfunction with a preserved or reduced ejection fraction. In this stage, the body response will promote aberrant vascular remodeling and the installment of persistent pulmonary hypertension [17,18].

The dimension of the airway at the level of the pharynx is determined by the interaction between two opposing forces. One force is represented by the negative pressure gradient by inspiration at the level of the pharynx, the other one is represented by the intrinsic muscles of the pharynx promoting dilation. The activity of these mechanisms is regulated by chemical stimulus, reflexes, and the state of consciousness. The nervous mechanisms are active in the state of awakening and the muscular activity decreases in the states of deep sleep, diminishing the pharyngeal airway [19].

The muscles involved in maintaining the caliber of the pharynx are genioglossus, geniohyoid, levator, and tensor vela palatini [20].

In the case of patients with OSA, the pharyngeal airway collapses under positive pressure [21]. One of the most recent factors analyzed is the content of fatty tissue in the walls of the pharynx with a positive correlation between this and the risk of sleep apnea [22].

Obstructive episodes during sleep phases in children present an inverted pattern compared with adults; in pediatric cases, the majority of obstructive episodes occur during REM sleep, whereas in adults they appear in non-REM stages of sleep [23,24].

There are two most common phenotypes: OSA with adenoid hypertrophy without obesity and OSA with adenoid hypertrophy and obesity [20].

The most frequent symptom of OSA is a lack of sleep with drowsiness during the daytime. In some of the studies, this symptom was generally named a lack of energy or generalized fatigability [25,26,27]. In addition, notably, excessive daytime sleepiness is less commonly encountered in females along with snoring or witnessed apnea [28].

Various questionnaires were developed for evaluating the risk of OSA like the Berlin questionnaire [29], available for primary care providers, or the STOP-Bang questionnaire [30], used for screening and stratifying the patients according to their scores [31]. The Epworth scale is used for clinical surveillance and research, but with a reduced sensibility for OSA [32]. Recently, Walker NA, et al. proved that the test-retest reliability of the Epworth scale was substantial and in line with other clinical measures, including polysomnographic determination of the AHI [33]. There are few clinical signs specific to OSA like obesity [34,35,36,37].

A recent study from Wroclaw analyzed the possible correlation between bruxism and different types of OSA phenotypes, bruxism incidence was higher in the body position-related OSA phenotypes [38].

Investigations for OSA are recommended for any patient presenting fatigability, day sleepiness, and restless sleep without other causes. Snoring is mainly observed by the partner along with obesity [39].

The gold standard for diagnosing OSA is a polysomnography monitoring both sleep and respiratory parameters [40]. A standard polysomnography recording includes: the measurement of nasal airflow, measuring the respiratory effort at the thoracic and abdominal level, pulse oximetry, snoring recording by a microphone, sleep stages using encephalography, EKG, and limb movement. This test is complex and difficult for the patient.

Newer devices enable sleep apnea testing in outpatient settings, but this method reduces the number of variables recorded and the sensors are attached by the patient [41,42].

The severity of OSA is determined with the aid of AHI or using the RDI (Respiratory Disturbance Index) during polysomnography or the REI (Respiratory Event Index) for home-testing devices. An AHI value below 5/h is normal, below 15/h indicates a mild OSA, below 30/h a moderate OSA, and above 30/h underlines a severe OSA.

Obstructive apnea is defined as a decrease in the respiratory effort of more than 90% compared with the basal values recorded beforehand, for a period longer than 10 s, along with increased inspiratory effort at the level of the thorax or abdomen.

Hypopnea is defined as a decreased respiratory effort of more than 30% from the baseline value, for more than 10 seconds, along with a 3% desaturation according to the AASM or a 4% desaturation according to the CMS.

Another recorded parameter is RERA (Respiratory Effort-Related Arousal) focusing on the connection between apnea and the micro-arousal of the patient [43].

However, polysomnography cannot underline the site of obstruction and needs to be combined with nasal endoscopy and imaging modalities: X-ray cephalometry, sleep fluoroscopy, CT scans, and MRI [44,45]. In the last two decades, drug-induced sleep endoscopy has gained great popularity for assessing the sites and patterns of upper airway collapse in patients with OSA [46].

Cephalometry records the following variables: SNA, sella to nasion foe sagittal positioning of the maxillary; SNB, sella to nasion for the sagittal positioning of the mandible; PAS, posterior airway space; MPH, mandibular plane to hyoid bone [47,48,49,50].

We aim to analyze if the cephalometric parameters could be predictors of OSA using the data from a group of Romanian patients.

## 2. Materials and Methods

We analyzed the correlations between morphology, the functional properties of upper airways, and the severity of sleep apnea in Romanian patients. To our knowledge, this is one of the first studies gathering a large group of Romanian individuals with OSA focusing on cephalometric variables. This is due to the fact the development of sleep clinics in Romania is slow and such facilities require a high level of equipment and specialized trained personnel. Another aspect is the requirement to gather a study group numerous enough in order to attain statistical power. This is difficult in parts of Eastern Europe, due to social beliefs that snoring is something of natural development in aging individuals, and even due to limited access to high-quality healthcare services.

The inclusion criteria were: (a) patients diagnosed with OSA; (b) complete records of the patient data with polysomnography and a lateral plain X-ray; (c) signed informed consent for using the patient data in retrospective studies anonymously. The exclusion criteria were: (a) underaged patients, pregnant women, and patients with neoplasia; (b) incomplete records of the patient’s data; (c) no informed consent. Cephalometry on a lateral cranial plain X-ray was more frequently used in maxillofacial clinics, but we trained our ENT personnel to perform such measurements in view of the current study for correlating the general cephalometric parameters with the frequency of the apnea episodes. The cephalometric variables analyzed were: SNA, sella to nasion foe sagittal positioning of the maxillary; SNB, sella to nasion for the sagittal positioning of the mandible; PAS, posterior airway space; MPH, mandibular plane to the hyoid bone.

We compared our statistical results with those published in various other populations to identify the specific aspects of our study group.

Nonetheless, we proposed some variables that can enable the prognosis of future apnea development and preemptive measures, as general practitioners and healthcare systems are already crowded by underdiagnosed cases of OSA, the new silent killer pathology.

The Null Hypothesis (H0): we assume that there is no positive correlation between the cephalometric variables on the lateral plate X-ray and the AHI obtained through polysomnography.

The Alternative Hypothesis (H1): there is a positive statistical correlation between the modified cephalometric variables obtained on the lateral plain X-ray and the AHI obtained through polysomnography.

This was a cross-sectional, observational, analytic, retrospective study. Our study group consisted of patients presenting OSA, diagnosed through polysomnography and the associated ENT pathology.

The study group comprised 152 patients with OSA caused by ENT pathology: 138 males (90.79%) and 14 females (9.21%). The ages ranged from 48 to 70 years and the age group distribution was: 75 patients between 40–49 years (59%), 70 patients between 50–59 years (46%), and 7 patients more than 60 years (5%).

Regarding the distribution of height in the study group, the data ranged from 1.54 m to 1.96 m, with a median value of 1.753 m, and a standard deviation of 0.074 m. The weight of the patients was between 62 and 128 kg, with a medium value of 89.34 kg and a standard deviation of 13.38 kg.

The clinical exam enabled the recording of various variables: the presence of nasal septum deviation, hypertrophy of inferior nasal turbinate, tonsillar morphology, webbing index, uvula morphology, Mallampati score, and tongue base morphology to underline the correlation between the severity of OSA and these variables.

## 3. Results

Given the AHI values, the study group was distributed as follows:-Mild OSA in 45 cases, with a medium value of 11.96, SD of 1.729, and 14.46% variability;-Moderate OSA in 59 cases, with a medium value of 20.54, SD of 4.269, and 20.78% variability;-Severe OSA in 48 cases, with a medium value of 50.49, SD of 17.65, and 34.96% variability;-Nasal septum deviation in any degree of severity was recorded in 80% of the cases. In addition, 67% of the cases were associated with inferior nasal turbinate hypertrophy.

The tonsillar morphology is distributed as follows: 7.23% (11 cases) presented massive tonsillar hypertrophy, 26.97% (41 cases) presented hypertrophic tonsils, 37.5% (57 cases) with normal tonsils, and interestingly, 28.28% (43 cases) with removed tonsils.

Webbing index values recorded in our study group were: 18.42% (28 cases) with Webbing 1 index, 34.21% (52 cases) with Webbing 2 index, and 47.36% (72 cases) were classified as the Webbing 3 type.

The Webbing index is important for classifying the positioning and dimensions of the posterior pharyngeal pillars and the soft palate; values lower than 5 mm are considered Webbing 1, Webbing 2 is for values lower than 10 mm, and cases with values higher than 10 mm are labeled Webbing 3 [51].

The different types of uvula morphology were classified as follows: 24.34% (37 cases) with long uvula, 49.34% (75 cases) with wide uvula, and 26.31% (40 cases) with normal uvula conformation.

Tongue morphology was classified as hypertrophic in 18.42%, moderately enlarged in 41.44% of the cases, and 40.13% of the cases were considered normal.

This results in the following distribution of the cases according to the Mallampati scale: type 1 in 72,36% of the cases, type 2 in 25.65% of the cases, and 1.97% of the cases were considered type 3.

Table 1 contains descriptive statistics of the cephalometric variables recorded in the study group.

BMI distribution in the study group: 19 patients (12.5%) were normal, 83 patients (54.60%) were classified as overweight, and 50 patients (32.89%) presented obesity.

We applied the Mann-Whitney U test for the variables recorded in our study group. The comparison between the subgroups with or without nasal septum deviation, taking into consideration the AHI, revealed a lack of statistical significance. The same situation was recorded when comparing the subgroups with or without inferior nasal turbinate hypertrophy.

In order to analyze more than two subgroups, we applied the nonparametric Kruskal-Wallis test; therefore, by comparing the subgroups with different types of tonsillar morphology, we recorded no statistically significant difference. The same situation was encountered when comparing the subgroups with different Webbing classes. However, we discovered a *p* = 0.0025 when analyzing the statistical correlation between the AHI and the different types of uvulae, Figure 1. Although the initial observations targeted a positive correlation between the AHI and the Mallampati classification and also the different types of tongue morphology, the Kruskal-Wallis test excluded this.

Regarding the age group distribution and the AHI, the Pearson correlation revealed no positive statistical correlation. There is a positive statistical correlation between the AHI and the values of MPH recorded, r = 0.1637, *p* < 0.05, Figure 2. This was not the case with the possible correlation between the AHI and PAS. The other variable correlating with the AHI was the length of the uvula with r = 0.2045 and *p* < 0.05, Figure 3.

## 4. Discussion

This study focused on underlining the possible correlation between polysomnography and cephalometric variables measured on plain lateral X-rays.

Polysomnography is the gold standard in OSA management, but there are somewhat simpler cephalometric variables that could predict and diagnose OSA.

Our results confirm that the data available in other studies regarding the sex distribution of OSA patients asking for treatment favors males from 8:1 to 10:1 [51,52].

Regarding the cephalometric measurements, we encountered a positive correlation between increased MPH and OSA severity, which was also confirmed in other studies [53]; however, specific to our study group is the lack of correlation between the PAS and the severity of OSA.

A recent study from Italy concluded that the mandibular length was the only variable with a statistical correlation with the apnea-hypopnea index [54].

Furthermore, between the length of the uvula and morphology, there is a correlation increasing with the AHI, like in other published studies [55,56].

Shigeta Y. et al. compared the differences in the soft palate and airway length between OSA and normal controls. In this study, there was a significant correlation between a longer soft palate and OSA, according to data obtained through CT scans [57].

Surprisingly in our study, there was no positive statistical correlation between nasal septum deviation and hypertrophy of the turbinates; the reason behind this situation is probably the high presence of these pathologies that are neglected for long periods of time in the general population of Romania [58,59,60].

Moreover, in a very recent study on 42 patients, the severity of AHI scores showed a negative correlation with cephalometric variables such as gonial angle and nasopharynx space, which is congruent with our finding that the PAS is not influencing OSA severity [61].

Tonsillar hypertrophy is a common risk factor for OSA pediatric cases, but our study group contained adults and this factor was not sustained by a positive statistical correlation, requiring further investigations on a larger group when possible [62,63,64].

Surprisingly in our study, there was no statistical correlation between OSA and the Mallampati scale like in other anesthesia studies [65].

Tongue hypertrophy is considered a possible cause of OSA, encountered in Down syndrome for example, but in our study, there was no statistical correlation although macroglossia could be decreasing the adherence to using a CPAP device [66,67,68].

The limitations of the present study are a reduced female sample, which is due to the small number of females seeking help with OSA, and the use of 2D plain X-rays instead of newer imaging modalities.

Another study from Romania offered a surprising possible explanation of the variability of the number of female patients with OSA in our country. Women with OSA are presenting with complaints that send us to other disorders than sleep apnea (depressive moods, insomnia, and morning headaches) [69].

The increased BMI is also a risk factor for OSA, we could not ascertain this situation because obesity tends to become endemic in Romania and our study group failed to have a normal distribution of this variable [70,71]. Nonetheless, this finding is similar to a previous retrospective study from Italy, which found no correlation between BMI and AHI or between BMI and SaO_2_ [72].

## 5. Conclusions

To our knowledge, this is one of the first studies focusing on cephalometric variables in the Romanian population and the correlations between these variables with the severity of OSA. This pathology presents some traits specific to the Romanian population, such as a lack of correlation between the classic parameters like BMI, tonsillar hypertrophy, and the Mallampati scale. This is because obesity has become endemic in Romania and the study group distribution was not normal. Another interesting aspect was that the cephalometric variables, such as MPH and length of the uvula, correlated with the AHI severity, which opens up the possibility of surveying for OSA cases in settings where access to high-quality healthcare services is difficult. This study is the starting point for analyzing the correlation between different risk factors and OSA in the Romanian population, and the subsequent data will be used in further studies on a larger patient group.

## Figures and Tables

**Figure 1 jcm-11-05347-f001:**
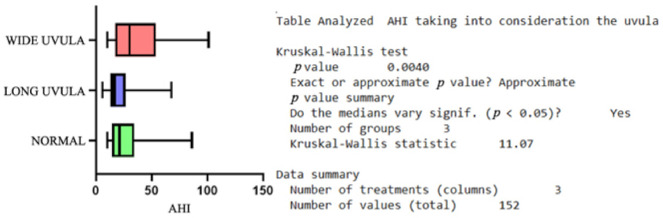
Distribution of AHI inside the study group taking into consideration the uvula morphology.

**Figure 2 jcm-11-05347-f002:**
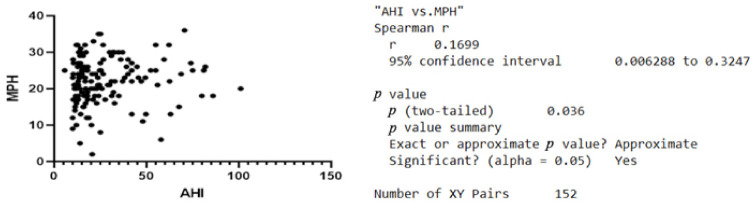
Graphic representation of the correlation between MPH values and AHI.

**Figure 3 jcm-11-05347-f003:**
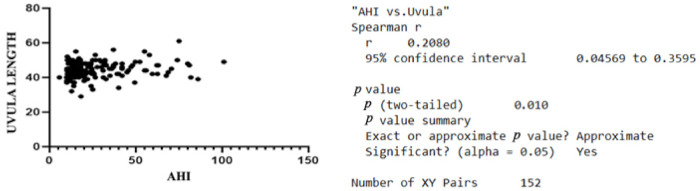
Graphic representation of the correlation between AHI and uvula length.

**Table 1 jcm-11-05347-t001:** Descriptive statistics of the main cephalometric variables in the study group.

Variable	Minimum	Maximum	Median	SD	Variability
MPH	2	36	21.85	6.367	29.14%
PAS	1	18	10.49	3.152	30.04%
Uvula length	29	61	44.32	4.758	10.74%

## Data Availability

Any data produced in this study are available from the corresponding author upon request.

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
