# Peer review of "Correlations between Morphology, the Functional Properties of Upper Airways, and the Severity of Sleep Apnea"

_jcm, 2022, doi:10.3390/jcm11185347_

Round 1
Reviewer 1 Report
line 38 Adeno-tonsillar hypertrophy is the most frequent reason for paediatric sleep disordered breathing, however the spectrum of aetiologies is very large. Syndromic cranio facial malformations and metabolic disorders are often associated with a very high risk of relevant sleep disordered breathing.
Möller A. Atemstörungen im Schlaf [Sleep-disordered Breathing]. Pneumologie. 2020 Apr;74(4):222-229. German. doi: 10.1055/a-0977-6236. Epub 2020 Apr 9. PMID: 32274782.
Moreover it is crucial that clinicians understand what underpins the considerable phenotypic variance in pediatric obstructive sleep apnea syndrome.
Tan HL, Kaditis AG. Phenotypic variance in pediatric obstructive sleep apnea. Pediatr Pulmonol. 2021 Jun;56(6):1754-1762. doi: 10.1002/ppul.25309. Epub 2021 Feb 16. PMID: 33543838.
line 44 Analyze and compare the results with the recent paper
Bertuzzi F, Santagostini A, Pollis M, Meola F, Segù M. The Interaction of Craniofacial Morphology and Body Mass Index in Obstructive Sleep Apnea. Dent J (Basel). 2022 Jul 19;10(7):136. doi: 10.3390/dj10070136. PMID: 35877410; PMCID: PMC9317640.
line 84 the reference 17 is only for children. The mentioned classification doesn't exist.
line 85 the daytime sleepiness is a symptom very different in male and female. To describe it use more recent references like
Geer JH, Hilbert J. Gender Issues in Obstructive Sleep Apnea. Yale J Biol Med. 2021 Sep 30;94(3):487-496. PMID: 34602886; PMCID: PMC8461585.
line 90 STOP-Bang Questionnaire is a method to Screen for Obstructive Sleep Apnea
line 91 Analyze better the use of the Epworth Sleepness Scale
Walker NA, Sunderram J, Zhang P, Lu SE, Scharf MT. Clinical utility of the Epworth sleepiness scale. Sleep Breath. 2020 Dec;24(4):1759-1765. doi: 10.1007/s11325-020-02015-2. Epub 2020 Jan 14. PMID: 31938991.
Chung F, Abdullah HR, Liao P. STOP-Bang Questionnaire: A Practical Approach to Screen for Obstructive Sleep Apnea. Chest. 2016 Mar;149(3):631-8. doi: 10.1378/chest.15-0903. Epub 2016 Jan 12. PMID: 26378880.
line 92 ... not only obesity, it depends on the phenotype
Never you cite and analyze the symptom bruxism.
line 115 needs??? in which guideline? You don't mention the sleep endoscopy to identify the site of obstruction.
At the end. of the introduction the aim of the study is absent.
Materials and methods How recruited the patients? Did they signed a. consensus form?
line 221 Precise that this ratio regarding patients asking for a treatment and not in the population
Results Try to reanalyze your data considering BMI
Reviewer 2 Report
The authors presented an interesting paper about morphology of upper airway and OSA.
The paper must be incresed in the following sections:
1 List the inclusion and exclusion criteria;
2 Materials and methods: specify the cephalometric variables used and all correlations analysed;
3 Modify the description of table 1 (the BMI index is not a cephalometric variable);
4 For a complete interpretation of the data, please attach in addition to the graphs, the tables of the statistical analysis (including the p value);
5 Discussion section
-the paragraph” although macroglossia is decreasing.”: the bibliographic citations 55/ 56 do not support this assertion, the citation 57 only partially although it is a case report. Please explain better what you claim by using appropriate citations;
-the paragraph” The increased BMI is also.. because obesity tends to become endemic”: please insert a citation that support your claim;
6 Explain the limitation of the study (female sample, 2d imaging...);
7 The discussion section needs an improvement, following my previous comments. I would suggest the following references, but there are also more to use:
- Craniofacial morphology in patients with obstructive sleep apnea: cephalometric evaluation (DOI : 10.1016/j.bjorl.2020.05.026).
- Demographic, clinical and polysomnographic differences between men and women (PMID: 20695359).
- Soft palate length and upper airway relationship in OSA and non-OSA subjects (PMID: 23734544)
- Correlation between Apnea Severity and Sagittal Cephalometric Features in a Population of Patients with Polysomnographically Diagnosed Obstructive Sleep Apnea ( DOI: 10.3390/jcm11154572)
- Correlation between body mass index and obstructive sleep apnea severity indexes - A retrospective study (DOI : 10.1016/j.amjoto.2018.03.026).
